# Introducing µGUIDE for quantitative imaging via generalized uncertainty-driven inference using deep learning

**Maëliss Jallais[1,2]\*, Marco Palombo[1,2]\***

[1]Cardiff University Brain Research Imaging Centre (CUBRIC), Cardiff University, Cardiff, United Kingdom; [2]School of Computer Science and Informatics, Cardiff University, Cardiff, United Kingdom

## eLife Assessment

The authors proposed an **important** novel deep-learning framework to estimate posterior distributions of tissue microstructure parameters. The method shows superior performance to conventional Bayesian approaches and there is **convincing** evidence for generalizing the method to use data from different protocol acquisitions and work with models of varying complexity.

**\*For correspondence:**
jallaism@cardiff.ac.uk (MJ);
palombom@cardiff.ac.uk (MP)

**Competing interest:** The authors declare that no competing interests exist.

**Abstract** This work proposes µGUIDE: a general Bayesian framework to estimate posterior distributions of tissue microstructure parameters from any given biophysical model or signal representation, with exemplar demonstration in diffusion-weighted magnetic resonance imaging. Harnessing a new deep learning architecture for automatic signal feature selection combined with simulation-based inference and efficient sampling of the posterior distributions, µGUIDE bypasses the high computational and time cost of conventional Bayesian approaches and does not rely on acquisition constraints to define model-specific summary statistics. The obtained posterior distributions allow to highlight degeneracies present in the model definition and quantify the uncertainty and ambiguity of the estimated parameters.

## Introduction

Diffusion-weighted magnetic resonance imaging (dMRI) is a promising technique for characterizing brain microstructure in vivo using a paradigm called microstructure imaging (*Novikov et al., 2019*; *Alexander et al., 2019*; *Jelescu et al., 2020*). Traditionally, microstructure imaging quantifies histologically meaningful features of brain microstructure by fitting a forward (biophysical) model voxel-wise to the set of signals obtained from images acquired with different sensitivities, yielding model parameter maps (*Alexander et al., 2019*).

Most commonly used techniques rely on a non-linear curve fitting of the signal and return the optimal solution, that is the best parameters guess of the fitting procedure. However, this may hide model degeneracy, that is all the other possible estimates that could explain the observed signal equally well (*Jelescu et al., 2016*). Another crucial consideration in model fitting is accounting for the uncertainty in parameter estimates. This uncertainty serves various purposes, including assessing result confidence (*Jones, 2003*), quantifying noise effects (*Behrens et al., 2003*), or assisting in experimental design (*Alexander, 2008*).

Instead of attempting to remove the degeneracies, which has been the focus of a large number of studies (*Palombo et al., 2023*; *de Almeida Martins et al., 2021*; *Slator et al., 2021*; *Jelescu et al., 2022*; *Warner et al., 2023*; *Uhl et al., 2024*; *Mougel et al., 2024*; *Olesen et al., 2022*; *Palombo*

*et al., 2020; Howard et al., 2022; Jones et al., 2018; Vincent et al., 2020; Henriques et al., 2021; Afzali et al., 2021; Lampinen et al., 2023; Zhang et al., 2012; Guerreri et al., 2023; Gyori et al., 2022; Novikov et al., 2018*), we propose to highlight them and present all the possible parameter values that could explain an observed signal, providing users with more information to make more confident and explainable use of the inference results.

Posterior distributions are powerful tools to characterize all the possible parameter estimations that could explain an observed measurement, their uncertainty, and existing model degeneracy (*Box and Tiao, 2011*). Bayesian inference allows for the estimation of these posterior distributions, traditionally approximating them using numerical methods, such as Markov-Chain-Monte-Carlo (MCMC) (*Metropolis et al., 1953*). In quantitative MRI, these methods have been used for example to estimate brain connectivity (*Behrens et al., 2003*), optimize imaging protocols (*Alexander, 2008*), or infer crossing fibres by combining multiple spatial resolutions (*Sotiropoulos et al., 2013*). However, these classical Bayesian inference methods are computationally expensive and time consuming. They also often require adjustments and tuning specific to each biophysical model (*Harms and Roebroeck, 2018*).

Harnessing a new deep learning architecture for automatic signal feature selection and efficient sampling of the posterior distributions using Simulation-Based Inference (SBI) (*Cranmer et al., 2020; Lueckmann et al., 2017; Papamakarios et al., 2019*), here we propose µGUIDE: a general Bayesian framework to estimate posterior distributions of tissue microstructure parameters from any given biophysical model/signal representation. µGUIDE extends and generalizes previous work (*Jallais et al., 2022*) to any forward model and without acquisition constraints, providing fast estimations of posterior distributions voxel-wise. We demonstrate µGUIDE using numerical simulations on three biophysical models of increasing complexity and degeneracy and compare the obtained estimates with existing methods, including the classical MCMC approach. We then apply the proposed

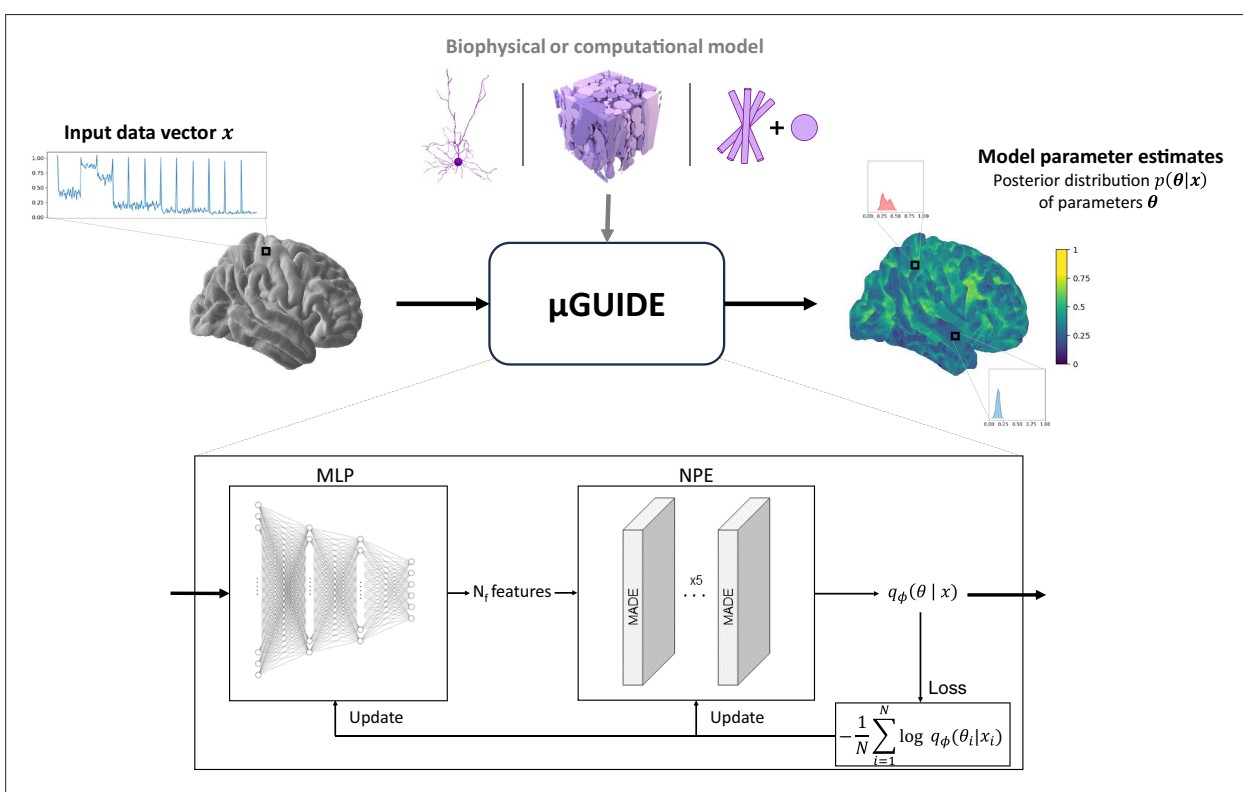

**Figure 1.** µGUIDE framework. µGUIDE takes as input an observed data vector and relies on the definition of a biophysical or computational model (*Ascoli et al., 2007; Callaghan et al., 2020; Jelescu et al., 2020*). It outputs a posterior distribution of the model parameters. Based on a Simulation-Based Inference (SBI) framework, it combines a Multi-Layer Perceptron (MLP) with three layers and a Neural Posterior Estimator (NPE). The MLP learns a low-dimensional representation of $\boldsymbol{x}$, based on a small number of features ($N_f$), that can be either defined a priori or determined empirically during training. The MLP is trained simultaneously with the NPE, leading to the extraction of the optimal features that minimize the bias and uncertainty of $p(\boldsymbol{\theta}|\boldsymbol{x})$.

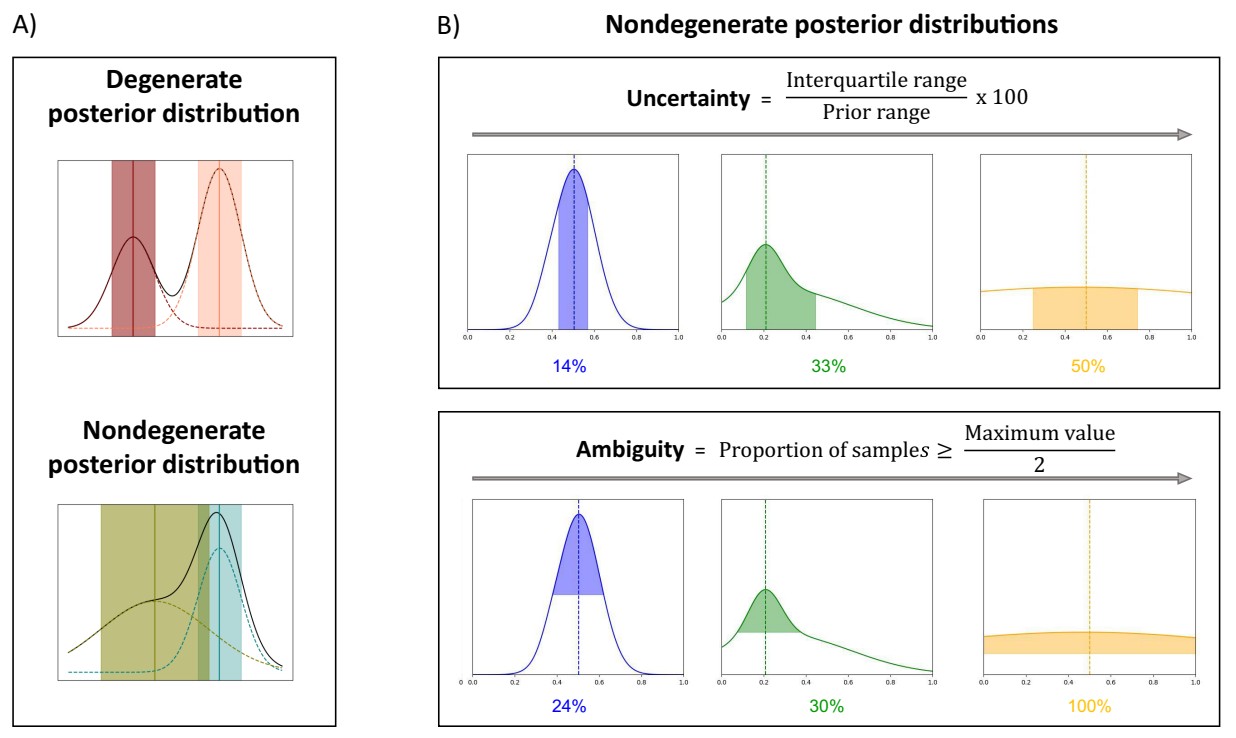

**Figure 2.** µGUIDE summarizes information contained in the estimated posterior distributions. (**A**) Examples of degenerate and non-degenerate posterior distributions. Two Gaussian distributions are fitted to the obtained posterior distribution, where the means and standard deviations are represented by the vertical lines and shaded areas. A voxel is considered as degenerate if the derivative of the fitted Gaussian distributions changes signs more than once (i.e. multiple local maxima), and if the two Gaussian distributions are not overlapping (the distance between the two Gaussian means is inferior to the sum of their standard deviations). (**B**) Presentation of the measures introduced to quantify a posterior distribution on exemplar non-degenerate posterior distributions. Maximum A Posteriori (MAP) is the most likely parameter estimate (dashed vertical lines). Uncertainty measures the dispersion of the 50% most probable samples using the interquartile range, with respect to the prior range. Ambiguity measures the Full Width at Half Maximum (FWHM), in percentage with respect to the prior range.

framework to dMRI data acquired from healthy human volunteers and participants with epilepsy. µGUIDE framework is agnostic to the origin of the data and the details of the forward model, so we envision its usage and utility to perform Bayesian inference of model parameters also using data from other MRI modalities (e.g. relaxation MRI) and beyond.

## Results

### Framework overview

The full architecture of the proposed Bayesian framework, dubbed µGUIDE, is presented in *Figure 1*. µGUIDE allows to efficiently estimate full posterior distributions of tissue parameters. It is comprised of two modules that are optimized together to minimize the Kullback–Leibler divergence between the true posterior distribution and the estimated one for every parameters of a given forward model. The 'Neural Posterior Estimator' (NPE) module (*Papamakarios et al., 2017*) uses normalizing flows (*Papamakarios et al., 2021*) to approximate the posterior distribution, while the 'Multi-Layer Perceptron' (MLP) module is used to reduce the data dimensionality and ensure fast and robust convergence of the NPE module.

The full posterior distribution contains a lot of useful information. To summarize and easily visualize this information, we propose three measures that quantify the best estimates and the associated confidence levels, and a way to highlight degeneracy. The three measures are the Maximum A Posteriori (MAP), which corresponds to the most likely parameter estimate; an uncertainty measure, which quantifies the dispersion of the 50% most probable samples using the interquartile range, relative to the prior range; and an ambiguity measure, which measures the Full Width at Half Maximum

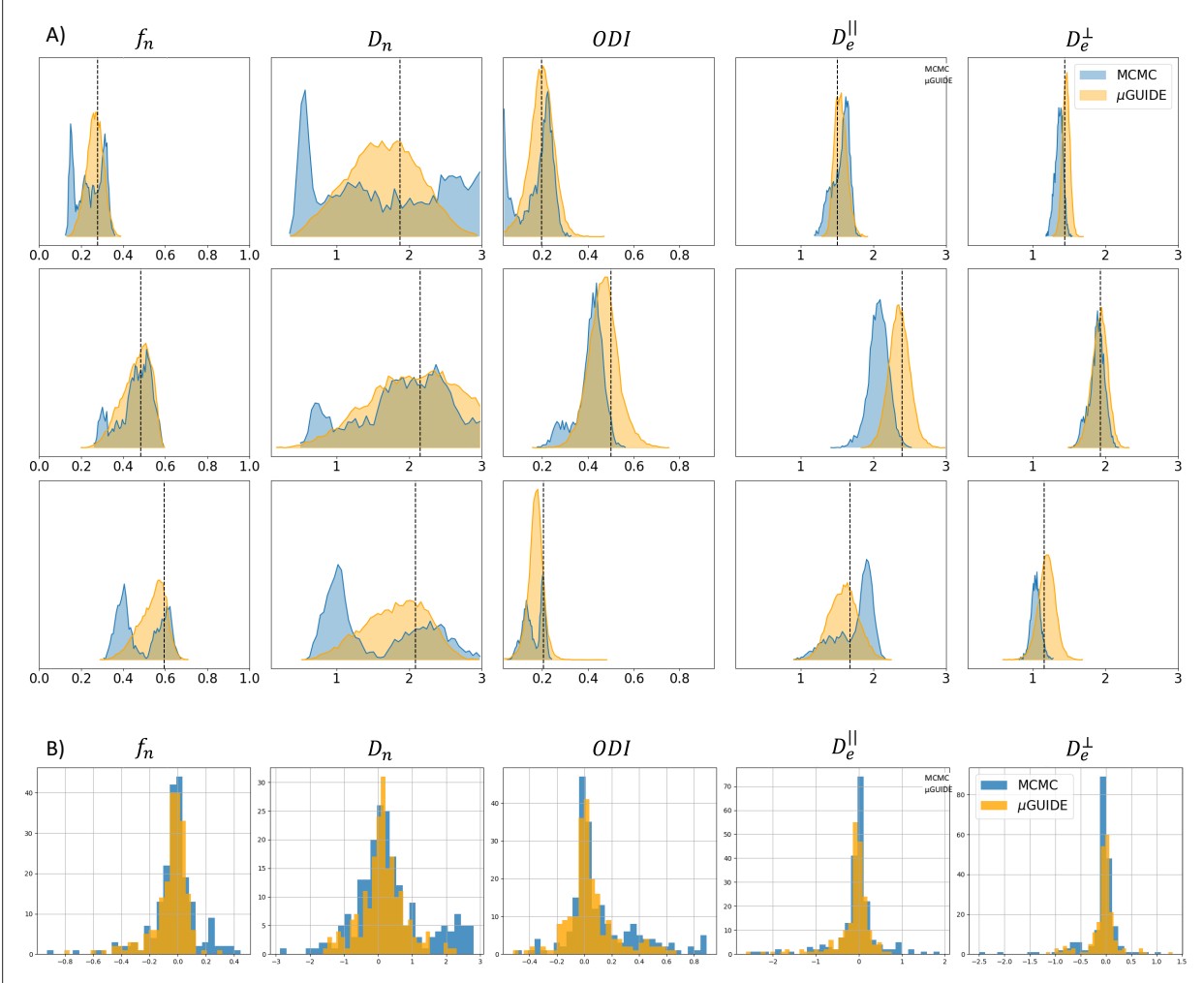

**Figure 3.** Comparison between µGUIDE and Markov-Chain-Monte-Carlo (MCMC). (**A**) Posterior distributions obtained using either µGUIDE or MCMC on three exemplar simulations with Model 2 (SM − $\mathbf{SNR = 50}$). Names of the model parameters are indicated in the titles of the panels. (**B**) Bias between the ground truth values used for simulating the diffusion signals, and the Maximum A Posteriori extracted from the posterior distributions using either µGUIDE or MCMC. Sharper and less biased posterior distributions are obtained using µGUIDE.

(FWHM), in percentage with respect to the prior range. *Figure 2* presents those measures on exemplar posterior distributions. We show exemplar applications of µGUIDE to three biophysical models of increasing complexity and degeneracy from the dMRI literature: Ball&Stick (*Behrens et al., 2003*) (Model 1); Standard Model (SM) (*Novikov et al., 2019*) (Model 2); and extended-SANDI (*Palombo et al., 2020*) (Model 3).

## Evaluation of µGUIDE on simulations

### Comparison with MCMC

We performed a comparison between the posterior distributions obtained using µGUIDE and MCMC, a classical Bayesian method. *Figure 3A* shows posterior distributions on three exemplar simulations with SNR = 50 using the Model 2 (SM), obtained with 15,000 samples. Sharper and less biased posterior estimations are obtained using µGUIDE. *Figure 3B* presents histograms for each model parameter of the bias between the ground truth value used to simulate a signal, and the MAP of the posterior distributions obtained with either µGUIDE or MCMC, on 200 simulations. Results indicate that the bias is similar or smaller using µGUIDE. Overall, µGUIDE posterior distributions are more accurate than the ones obtained with MCMC.

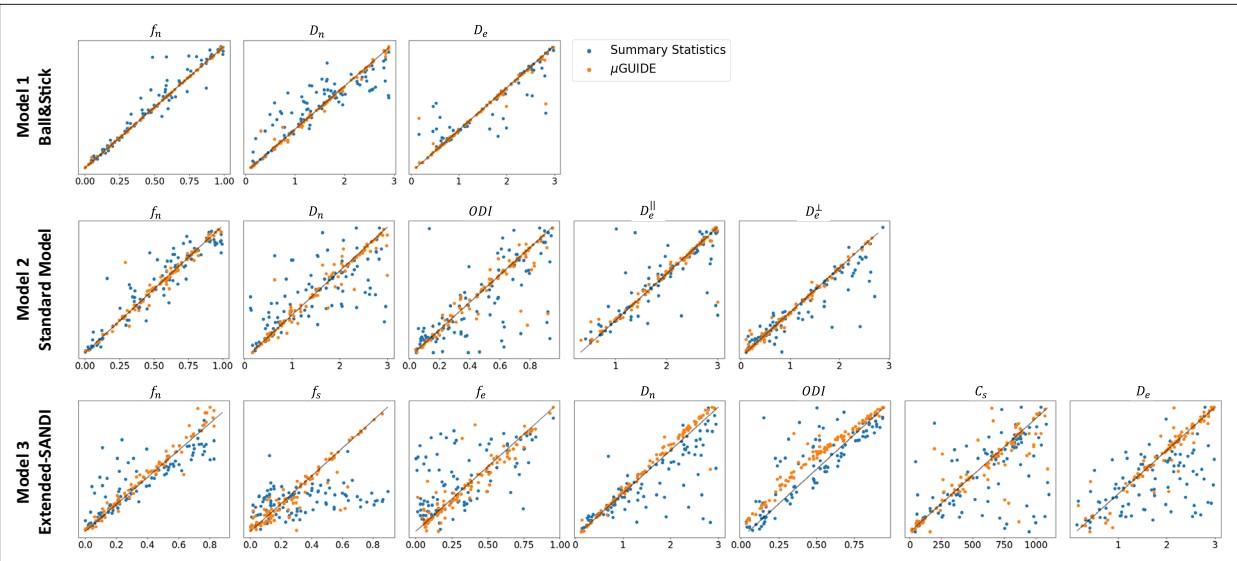

**Figure 4.** Fitting accuracy comparison between µGUIDE's Multi-Layer Perceptron (MLP)-extracted features and manually defined summary statistics. Maximum A Posterioris (MAPs) extracted from the posterior distributions versus ground truth parameters used for generating the signal for the three models. Orange points correspond to the MAPs obtained using MLP-extracted features (µGUIDE) and the blue ones to the MAPs with the manually defined summary statistics. Only the non-degenerate posterior distributions were kept. The summary statistics used in those three models are the direction-averaged signal for the Ball&Stick model, the LEMONADE system of equations (*Novikov et al., 2018*) for the Standard Model (SM), and the summary statistics defined in *Jallais et al., 2022* for the extended-SANDI model. Results are shown on 100 exemplar noise-free simulations with random parameter combinations. The optimal features extracted by the MLP allow to reduce the bias and variance of the obtained microstructure posterior distributions.

Moreover, it took on average 29.3 s to obtain the posterior distribution using MCMC on a GPU (NVIDIA GeForce GT 710) for one voxel, while it only took 0.02 s for µGUIDE. µGUIDE is about 1500 times faster than MCMC, which makes it more suitable for applying it on large datasets.

## The importance of feature selection

*Figure 4* shows the MAP extracted from the posterior distributions versus the ground truth parameters used to generate the diffusion signal with µGUIDE and manually defined summary statistics for the three models. Less biased MAPs with lower ambiguities and uncertainties are obtained with µGUIDE, indicating that the MLP allows for the extraction of additional information not contained in the summary statistics, helping to solve the inverse problem with higher accuracy and precision. µGUIDE generalizes the method developed in *Jallais et al., 2022* to make it applicable to any forward model and any acquisition protocol, while making the estimates more precise and accurate thanks to the automatic feature extraction.

## µGUIDE highlights degeneracies

*Figure 5* presents the posterior distributions of microstructure parameters for the three models obtained with µGUIDE on exemplar noise-free simulations. Blue curves correspond to non-degenerate posterior distributions, while the red ones present at least one degeneracy for one of the parameters. As the complexity of the model increases, degeneracy in the model definitions appear. This figure showcases µGUIDE ability to highlight degeneracy in the model parameter estimation.

*Tables 1 and 2* present the number of degenerate cases for each parameter in the three models, on 10,000 simulations. *Table 1* considers noise-free simulations and the training and estimations were performed on CPU. *Table 2* reports results on noisy simulations (Rician noise with SNR = 50), with training and testing performed on a GPU (NVIDIA GeForce RTX 4090). The time needed for the inference and to estimate the posterior distributions on 10,000 simulations, define if they are degenerate or not, and extract the MAP, uncertainty, and ambiguity are also reported. The more complex the model, the more degeneracies.

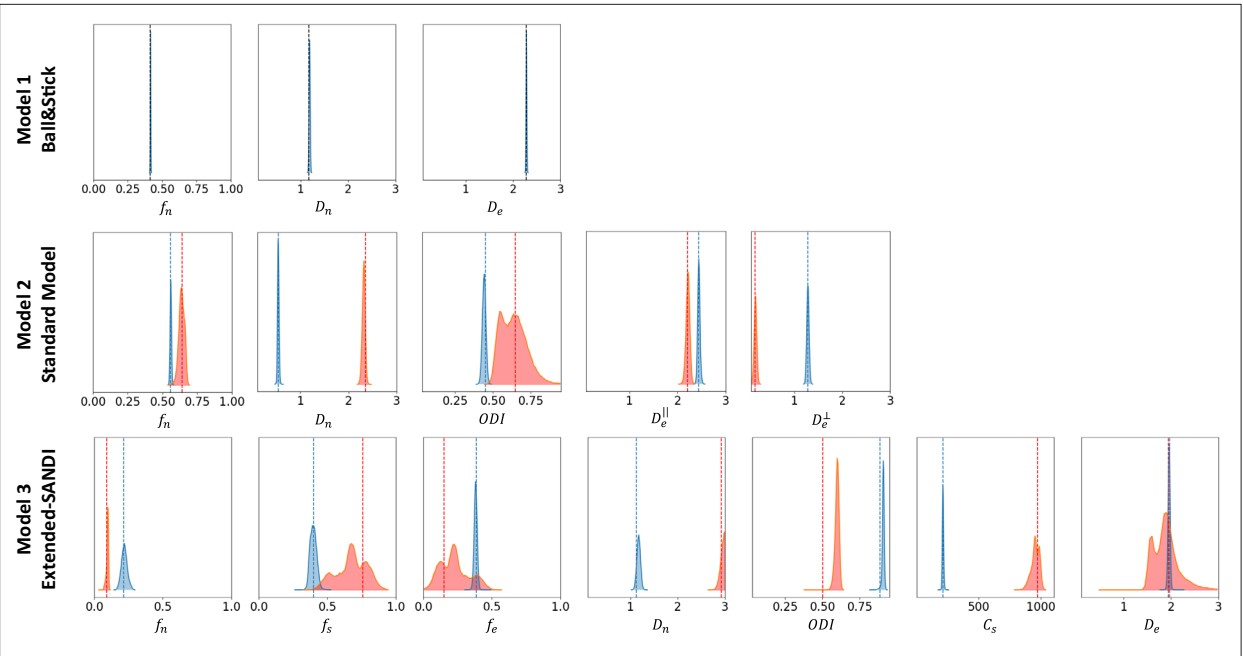

**Figure 5.** Exemplar posterior distributions of the microstructure parameters for the Ball&Stick, Standard Model (SM), and extended-SANDI models, obtained using µGUIDE on exemplar noise-free simulations. As the complexity of the model increases, degeneracies appear (red posterior distributions). µGUIDE allows to highlight those degeneracies present in the model definition.

## Application of µGUIDE to real data

After demonstrating that the proposed framework provides good estimates in the controlled case of simulations, we applied µGUIDE to both a healthy volunteer and a participant with epilepsy. The estimation of the posterior distributions is done independently for each voxel. To easily assess the values and the quality of the fitting, we are plotting the MAP, ambiguity, and uncertainty maps, but the full posterior distributions are stored and available for all the voxels. Voxels presenting a degeneracy are highlighted with a red dot.

### Healthy volunteer

We applied µGUIDE to a healthy volunteer, using the Ball&Stick, SM, and extended-SANDI models. *Figure 6* presents the parametric maps of an exemplar set of model parameters for each model, alongside their degeneracy, uncertainty, and ambiguity. The Ball&Stick model presents no degeneracy, the SM presents some degeneracy, mostly in voxels with high likelihood of partial voluming with cerebrospinal fluid and at the white matter–grey matter boundaries. The extended-SANDI model is the model showing the highest number of degenerate cases, mostly localized within the white matter areas characterized by complex microstructure, for example crossing fibres. This result is expected, as the complexity of the models increases, leading to more combinations of tissue parameters that can

**Table 1.** Number of degenerate cases per parameter on 10,000 noise-free simulations.
Training and estimations of the posterior distributions were performed on CPU. Time for training each model and time for estimating posterior distributions of 10,000 noise-free simulations, define if they are degenerate or not, and extract the Maximum A Posteriori (MAP), uncertainty, and ambiguity are also reported.

| Model (SNR = ∞) | Training time (CPU) | Fitting time (on 10,000 simulations) | Number of degeneracies (on 10,000 simulations) | | | | | | | |
|---|---|---|---|---|---|---|---|---|---|---|
| | | | $f_n$ | $D_n$ | $D_e^{\parallel}$ | ODI | $D_e^{\perp}$ | $f_s$ | $f_e$ | $C_s$ |
| Model 1: Ball&Stick | 11 min | 96 s | 0 | 0 | 0 | - | - | - | - | - |
| Model 2: Standard Model | 2h02 | 135 s | 4 | 34 | 23 | 3 | 8 | - | - | - |
| Model 3: extended-SANDI model | 2h02 | 1412 s | 205 | 4 | 260 | 57 | - | 1395 | 2571 | 1011 |

**Table 2.** Number of degenerate cases per parameter on 10,000 noisy simulations (Rician noise with SNR = 50). Training and estimations of the posterior distributions were performed using a GPU. Time for training each model and time for estimating posterior distributions of 10,000 noisy simulations, define if they are degenerate or not, and extract the Maximum A Posteriori (MAP), uncertainty, and ambiguity are also reported.

| Model (SNR = 50) | Training time (CPU) | Fitting time (on 10,000 simulations) | Number of degeneracies (on 10,000 simulations) | | | | | | | |
|---|---|---|---|---|---|---|---|---|---|---|
| | | | $f_n$ | $D_n$ | $D_e^{\parallel}$ | ODI | $D_e^{\perp}$ | $f_s$ | $f_e$ | $C_s$ |
| Model 1: Ball&Stick | 26 min | 79 s | 0 | 0 | 0 | - | - | - | - | - |
| Model 2: Standard Model | 42 min | 82 s | 75 | 71 | 117 | 109 | 29 | - | - | - |
| Model 3: Extended-SANDI model | 50 min | 238 s | 47 | 24 | 784 | 6 | - | 828 | 1047 | 56 |

explain an observed signal. Measures of ambiguity and uncertainty allow to quantify the confidence in the estimates and help interpreting the results.

### Participant with epilepsy

*Figure 7* demonstrates μGUIDE application to a participant with epilepsy, using the SM. Noteworthy, the axonal signal fraction estimates within the epileptic lesion show low uncertainty and ambiguity measures hence high confidence, while orientation dispersion index estimates show high uncertainty and ambiguity suggesting low confidence, cautioning the interpretation.

## Discussion

### Applicability of μGUIDE to multiple models

The μGUIDE framework offers the advantage of being easily applicable to various biophysical models and representations, thanks to its data-driven approach for data reduction. The need to manually define specific summary statistics that capture the relevant information for microstructure estimation from the multi-shell diffusion signal is removed. This also eliminates the acquisition constraints that were previously imposed by the summary statistics definition (*Jallais et al., 2022*). The extracted features contain additional information compared to the summary statistics (see Appendix 1), resulting in a notable reduction in bias (on average 5.2-fold lower), uncertainty (on average 2.6-fold lower), and ambiguity (on average 2.7-fold lower) in the estimated posterior distributions. Consequently, μGUIDE improves parameters estimation over current state-of-the-art methods (e.g. *Jallais et al., 2022*), showing for example reduced bias (on average 5.2-fold lower) and dispersion (on average 6.4-fold lower) on the MAP estimates for each of the three example models investigated (see *Figure 4*).

In this study, we presented applications of μGUIDE to brain microstructure estimation using three well-established biophysical models, with increased complexity: the Ball&Stick model, the SM, and an extended-SANDI model. However, our approach is not limited to brain tissue nor to diffusion-weighted MRI and can be extended to different organs by employing their respective acquisition encoding and forward models, such as NEXI for exchange estimates (*Jallais et al., 2024*), mcDESPOT for myelin water fraction mapping using quantitative MRI relaxation (*Deoni et al., 2008*), VERDICT in prostate imaging (*Panagiotaki et al., 2014*), or even adapted to different imaging modalities (e.g. electroencephalography and magnetoencephalography), where there is a way to link (via modelling or simulation) the observed signal to a set of parameters of interest. This versatility underscores the broad applicability of our proposed approach across various biological systems and imaging techniques.

It is important to note that μGUIDE is still a model-dependent method, meaning that the training process is based on the specific model being used. Additionally, the number of features extracted by the MLP needs to be predetermined. One way to determine the number of features is by matching it with the number of parameters being estimated. Alternatively, a dimensionality-reduction study using techniques like *t*-distributed stochastic neighbour embedding (*der Maaten and Hinton, 2008*) can be conducted to determine the optimal number of features.

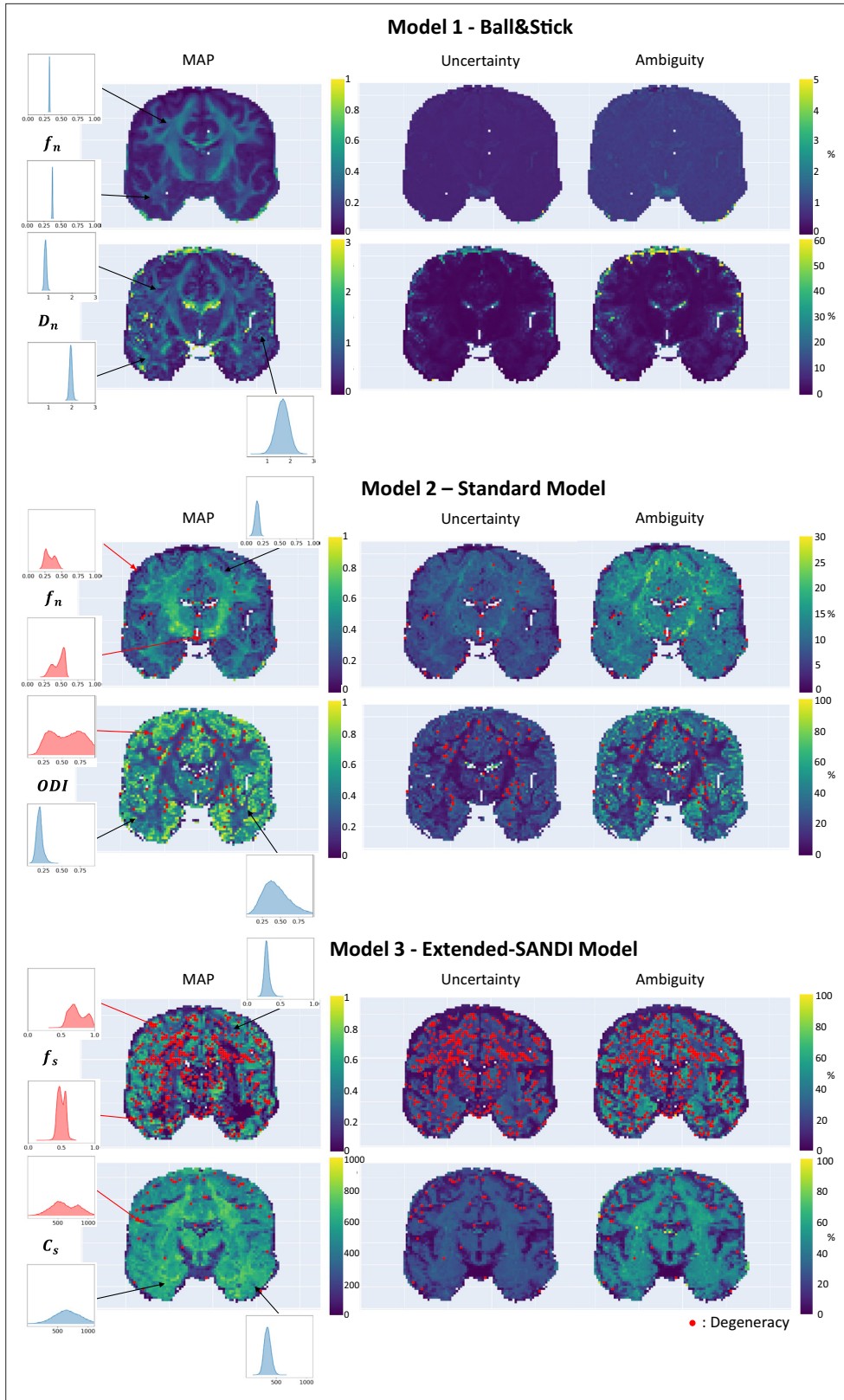

**Figure 6.** Parametric maps of the Ball&Stick (top), SM (middle) and extended-SANDI model (bottom), obtained using μGUIDE. Maximum A Posteriori (MAP), uncertainty and ambiguity measure maps are reported, overlayed with voxels considered degenerate (red dots).

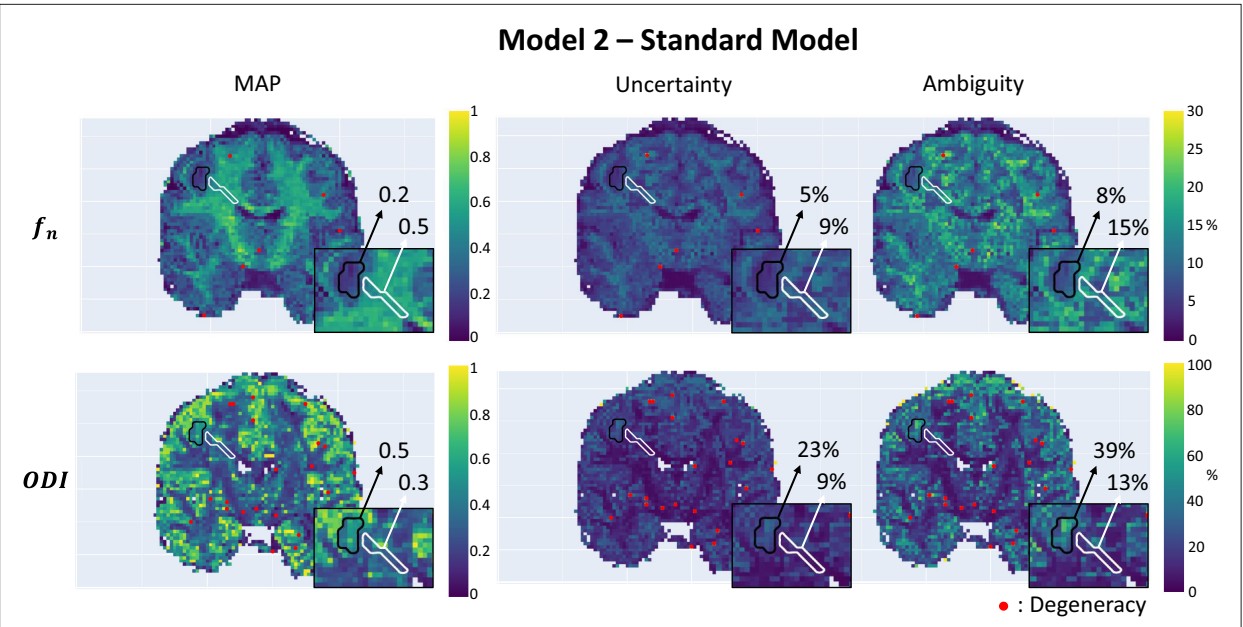

**Figure 7.** Parametric maps of a participant with epilepsy obtained using μGUIDE with the Standard Model (SM), superimposed with the grey matter (black) and white matter (white) lesions segmentation. Mean values of the Maximum A Posterior (MAP), uncertainty, and ambiguity measures are reported in the two regions of interest. Lower MAP values are obtained in the lesions for the axonal signal fraction and the orientation dispersion index compared to healthy tissue. Higher uncertainty and ambiguity ODI values are reported, suggesting less stable estimations.

## μGUIDE: an efficient framework for Bayesian inference

One notable advantage of μGUIDE is its amortized nature. With this approach, the training process is performed only once, and thereafter, the posterior estimations can be independently obtained for all voxels. This amortization enables efficient estimations of the posterior distributions. μGUIDE outperforms in terms of speed conventional Bayesian inference methods such as MCMC, showing a ~1500-fold acceleration. The time savings achieved with μGUIDE make it a highly efficient and practical tool for estimating posterior distributions in a timely manner.

This unlocks the possibility to process with Bayesian inference very large datasets in manageable time (e.g. approximately 6 months to process 10 k dMRI datasets) and to include Bayesian inference in iterative processes that require the repeated computation of the posterior distributions (e.g. dMRI acquisition optimization [*Alexander, 2008*]).

In the dMRI community, the use of SBI methods to characterize full posterior distributions as well as quantify the uncertainty in parameter estimations was first introduced in *Jallais et al., 2022* for a grey matter model. An application to crossing fibres has recently been proposed by *Karimi et al., 2024*. Those approaches use different density estimators. This work and *Jallais et al., 2022* rely on Masked Autoregressive Flows (MAFs [*Papamakarios et al., 2017*]), while the work by *Karimi et al., 2024* is based on Mixture Density Networks (MDNs [*Bishop, 1994*]). MAFs have been found to show superior performance compared to MDNs (*Gonçalves et al., 2020*; *Patron et al., 2022*).

## μGUIDE quantifies confidence to guide interpretation

Quantifying confidence in an estimate is of crucial importance. As demonstrated by our pathological example, changes in the tissue microstructure parameters can help clinicians decide which parameters are the most reliable and better interpret microstructure changes within diseased tissue. On large population studies, the quantified uncertainty can be taken into account when performing group statistics and to detect outliers.

Multiple approaches have been used to try and quantify this uncertainty. Gradient descent often provides a measure of confidence for each parameter estimate. Alternative approaches use the shape of the fitted tensor itself as a measure of uncertainty for the fibre direction (*Koch et al., 2002*; *Parker and Alexander, 2003*). Other methods also rely on bootstrapping techniques to estimate uncertainty. Repetition bootstrapping for example depends on repeated measurements of signal for each

gradient direction, but imply a long acquisition time and cost, and are prone to motion artifacts (*Lazar and Alexander, 2005*; *Jones, 2003*). In contrast, residual bootstrapping methods resample the residuals of a regression model. Yet, this approach is heavily dependent on the model and can lead to overfitting (*Whitcher et al., 2008*; *Chung et al., 2006*). In general, resampling methods can be problematic for sparse samples, as the bootstrapped samples tend to underestimate the true randomness of the distribution (*Kauermann et al., 2009*). We propose to quantify the confidence by estimating full posterior distributions, which also has the benefit of highlighting degeneracy. Model-fitting methods with different initializations, as done in for example *Jelescu et al., 2016*, also allow to highlight degeneracies. However, they only provide a partial description of the solution landscape, which can be interpreted as a partial posterior distribution. In contrast, Bayesian methods estimate the full posterior distributions, offering a more accurate and precise characterization of degeneracies and uncertainties. Hence, in this work we decided to use MCMC, a traditional Bayesian method, as benchmark.

Variance observed in the posterior distributions can be attributed to several factors. The presence of noise in the signal contributes to irreducible variance, decreasing the confidence in the estimates as the noise level increases (see Appendix 2). Another source of variance can arise from the choice of acquisition parameters. Different acquisitions may provide varying levels of confidence in the parameter estimates. Under-sampled acquisitions or inadequate b-shells may fail to capture essential information about a tissue microstructure, such as soma or neurite radii, resulting in inaccurate estimates.

µGUIDE can guide users in determining whether an acquisition is suitable for estimating parameters of a given model and vice versa, the variance and bias of the posterior distributions estimated with µGUIDE can be used to guide the optimization of the data acquisition to maximize accuracy and precision of the model parameters estimates.

The presence of degeneracy in the solution of the inverse problem is influenced by the complexity of the model being used and the lack of sufficient information in the data. In recent years, researchers have introduced increasingly sophisticated models to better represent the brain tissue, such as SANDI (*Palombo et al., 2020*), NEXI (*Jelescu et al., 2022*), and eSANDIX (*Olesen et al., 2022*), that take into account an increasing number of tissue features. By applying µGUIDE, it becomes possible to gain insights into the degree of degeneracy within a model and to assess the balance between model realism and the ability to accurately invert the problem. We have recently provided an example of such application for NEXI and SANDIX (*Jallais et al., 2024*).

## Summary

We propose a general Bayesian framework, dubbed µGUIDE, to efficiently estimate posterior distributions of tissue microstructure parameters. For any given acquisition and signal model/representation, µGUIDE improves parameters estimation and computational time over existing state-of-the-art methods. It allows to highlight degeneracy, and quantify confidence in the estimates, guiding results interpretation towards more confident and explainable diagnosis using modern deep learning. µGUIDE is not inherently limited to dMRI and microstructure imaging. We envision its usage and utility to perform efficient Bayesian inference also using data from any modality where there is a way to link (via modelling or simulation) the observed measurements to a set of parameters of interest.

## Methods
### Solving the inverse problem using Bayesian inference
The inference problem

We make the hypothesis that an observed dMRI signal $x_0$ can be explained (and generated) using a handful of relevant tissue microstructure parameters $\theta_0$, following the definition of a forward model:

$$x_0 = \mathcal{M}(\theta_0)$$

The objective is, given this observation $x_0$, to estimate the parameters $\theta_0$ that generated it.

Forward models are designed to mimic at best a given biophysical phenomenon, for some given time and scale (*Alexander, 2009*; *Yablonskiy and Sukstanskii, 2010*; *Jelescu and Budde, 2017*; *Novikov et al., 2019*; *Alexander et al., 2019*; *Jelescu et al., 2020*). As a consequence, forward models are injection functions (every biologically plausible $\theta_i$ generates exactly one signal $x_i$), but do

not always happen to be bijections, meaning that multiple $\theta_i$ can generate the same signal $x_i$. It can be impossible, based on biological considerations, to infer which solution $\theta_i$ best reflects the probed structure. We refer to these models as 'degenerate models'.

Point estimates algorithms, such as minimum least square or maximum likelihood estimation algorithms, allow to estimate one set of microstructure parameters that could explain an observed signal. In the case of degenerate models, the solution space can be multi-modal and those algorithms will hide possible solutions. When considering real-life acquisitions, that is noisy and/or under-sampled acquisitions, one also needs to consider the bias introduced with respect to the forward model, and the resulting variance in the estimates (**Jones, 2003**; **Behrens et al., 2003**).

We propose a new framework that allows for the estimation of full posterior distributions $p(\theta|x_0)$, that is all the probable parameters that could represent the underlying tissue, along with an uncertainty measure and the interdependency of parameters. These posteriors can help interpreting the obtained results and make more informed decisions.

## The Bayesian formalism

The posterior distribution can be defined using Bayes' theorem as follows:

$$p(\theta|x_0) = \frac{p(x_0|\theta)\,p(\theta)}{p(x_0)} \quad , \tag{1}$$

where $p(x_0|\theta)$ is the likelihood of the observed data point, $p(\theta)$ is the prior distribution defining our initial knowledge of the parameter values, and $p(x_0)$ is a normalizing constant, commonly referred to as the evidence of the data.

The evidence term is usually very hard to estimate, as it corresponds to all the possible realizations of $x_0$, that is $p(x_0) = \int_{all\ x_0} p(x_0|\theta)\,p(\theta)\,dx_0$. For simplification, methods usually estimate an unnormalized probability density function, that is

$$p(\theta|x_0) \propto p(x_0|\theta)p(\theta) \quad . \tag{2}$$

To approximate these posterior distributions, traditional methods rely on the estimation of the likelihood $p(x_0|\theta)$ of the observed data point $x_0$ via an analytic expression. This likelihood function corresponds to an integral over all possible trajectories through the latent space, that is $p(x_0|\theta) = \int p(x_0, z|\theta)\mathrm{d}z$, where $p(x_0, z|\theta)$ is the joint probability density of observed data $x_0$ and latent variables $z$. For forward models with large latent spaces, computing this integral explicitly becomes impractical. The likelihood function is then intractable, rendering these methods unusable (**Cranmer et al., 2020**). Models that do not admit a tractable likelihood are called *implicit models* (**Diggle and Gratton, 1984**).

To circumvent this issue, some techniques have been proposed to sample numerically from the likelihood function, such as MCMC (**Metropolis et al., 1953**). Another set of approaches proposes to train a conditional density estimator to learn a surrogate of the likelihood distribution (**Papamakarios et al., 2019**; **Lueckmann et al., 2019**), the likelihood ratio (**Cranmer et al., 2016**; **Gutmann et al., 2018**), or the posterior distribution (**Papamakarios and Murray, 2016**; **Lueckmann et al., 2017**; **Papamakarios et al., 2019**), allowing to greatly reduce computation times. These methods are dubbed likelihood-free inference or SBI methods (**Cranmer et al., 2020**; **Tejero-Cantero et al., 2020**). In particular, there has been a growing interest towards deep generative modelling approaches in the machine learning community (**Lueckmann et al., 2021**). They rely on specially tailored neural network architectures to approximate probability density functions from a set of examples. Normalizing flows (**Papamakarios et al., 2021**) are a particular class of such neural networks that have demonstrated promising results for SBI in different research fields (**Gonçalves et al., 2020**; **Greenberg et al., 2019**).

While this work focuses on the estimate of the posterior distribution using a conditional density estimator, we show a comparison with MCMC, which are commonly used methods in the community. We will therefore introduce this method in the following paragraph.

## Estimating the likelihood function

Well-established approaches for estimating the likelihood function are MCMC methods. These methods rely on a noise model to define the likelihood distribution, such as the Rician (**Panagiotaki**

et al., 2012) or Offset Gaussian models (Alexander, 2009). In this work, we will be using the Micro-structure Diffusion Toolbox to perform the MCMC computations (Harms and Roebroeck, 2018), which relies on the Offset Gaussian model. The log-likelihood function is then the following:

$$
\log\left(p(\boldsymbol{x}|\boldsymbol{\theta})\right) = -\frac{\sum_{i=1}^{m}\left(\boldsymbol{x}_i - \sqrt{\mathcal{M}(\boldsymbol{\theta})_i^2 + \sigma^2}\right)^2}{2\sigma^2} - m \cdot \log(\sigma\sqrt{2\pi}) \,,
\tag{3}
$$

where $\mathcal{M}(\boldsymbol{\theta})$ is the signal obtained using the biophysical model, $\mathcal{M}(\boldsymbol{\theta})_i$ is the $i$th measurement of the signal, $\sigma$ is the standard deviation of the Gaussian distributed noise, estimated from the reconstructed magnitude images (Dietrich et al., 2007), and $m$ is the number of observations in the dataset.

MCMC methods allow to obtain posterior distributions using Bayes' formula (Equation 2) with the previously defined likelihood function (Equation 3) and some prior distributions, which are usually uniform distributions defined on biologically plausible ranges. They generate a multi-dimensional chain of samples which is guaranteed to converge towards a stationary distribution, which approximates the posterior distribution (Metropolis et al., 1953).

The need to compute the signal following the forward model at each iteration makes these sampling methods computationally expensive and time consuming. In addition, they require some adjustments specific to each model, such as the choice of burn-in length, thinning, and the number of samples to store. Harms and Roebroeck, 2018 recommend to use the Adaptive Metropolis-Within-Gibbs (AMWG) algorithm for sampling dMRI models, initialized with a maximum likelihood estimator (MLE) obtained from non-linear optimization, with 100–200 samples for burn-in and no thinning. Authors notably investigated the use of starting from the MLE and thinning. They concluded that starting from the MLE allows to start in the stationary distribution of the Markov Chain, and has the advantage of removing salt- and pepper-like noise from the resulting mean and standard deviation maps. Their findings also indicate that thinning is unnecessary and inefficient, and they recommend using more samples instead. The recommended number of samples is model dependent. Authors recommendations can be found in their paper.

## Bypassing the likelihood function

An alternative method was proposed to overcome the challenges associated with approximating the likelihood function and the limitations of MCMC sampling algorithms. This approach involves directly approximating the posterior distribution by using a conditional density estimator, that is a family of conditional probability density function approximators denoted as $q_\phi(\boldsymbol{\theta}|\boldsymbol{x})$. These approximators are parameterized by $\phi$ and accept both the parameters $\boldsymbol{\theta}$ and the observation $\boldsymbol{x}$ as input arguments. Our posterior approximation is then obtained by minimizing its average Kullback–Leibler divergence with respect to the conditional density estimator for different choices of $\boldsymbol{x}$, as per Papamakarios and Murray, 2016:

$$
\min_{\phi} \quad \mathcal{L}(\phi) \quad \text{with} \quad \mathcal{L}(\phi) = \mathbb{E}_{\boldsymbol{x}\sim p(\boldsymbol{x})}\left[D_{\mathrm{KL}}(p(\boldsymbol{\theta}|\boldsymbol{x}) | q_\phi(\boldsymbol{\theta}|\boldsymbol{x}))\right] \,,
\tag{4}
$$

which can be rewritten as

$$
\begin{aligned}
\mathcal{L}(\phi) &= \int D_{\mathrm{KL}}(p(\boldsymbol{\theta}|\boldsymbol{x})\|q_\phi(\boldsymbol{\theta}|\boldsymbol{x}))p(\boldsymbol{x})\mathrm{d}\boldsymbol{x} \,, \\
&= -\iint \log\left(q_\phi(\boldsymbol{\theta}|\boldsymbol{x})\right)p(\boldsymbol{\theta}|\boldsymbol{x})p(\boldsymbol{x})\mathrm{d}\boldsymbol{\theta}\mathrm{d}\boldsymbol{x} + C \,, \\
&= -\iint \log\left(q_\phi(\boldsymbol{\theta}|\boldsymbol{x})\right)p(\boldsymbol{x}, \boldsymbol{\theta})\mathrm{d}\boldsymbol{\theta}\mathrm{d}\boldsymbol{x} + C \,, \\
&= -\mathbb{E}_{(\boldsymbol{x},\boldsymbol{\theta})\sim p(\boldsymbol{x},\boldsymbol{\theta})}\left[\log\left(q_\phi(\boldsymbol{\theta}|\boldsymbol{x})\right)\right] + C \,,
\end{aligned}
\tag{5}
$$

where $C$ is a constant that does not depend on $\phi$. Note that in practice we consider a $N$-sample Monte-Carlo approximation of the loss function:

$$
\mathcal{L}(\phi) \approx \mathcal{L}^N(\phi) = -\frac{1}{N}\sum_{i=1}^{N}\log\left(q_\phi(\boldsymbol{\theta}_i|\boldsymbol{x}_i)\right) \,,
\tag{6}
$$

where the $N$ data points $(\boldsymbol{\theta}_i, \boldsymbol{x}_i)$ are sampled from the joint distribution with $\boldsymbol{\theta}_i \sim p(\boldsymbol{\theta})$ and $\boldsymbol{x}_i \sim p(\boldsymbol{x}|\boldsymbol{\theta}_i)$. We can then use stochastic gradient descent to obtain a set of parameters $\phi$ which minimizes $\mathcal{L}^N$.

If the class of conditional density estimators is sufficiently expressive, it can be demonstrated that the minimizer of *Equation (6)* converges to $p(\boldsymbol{\theta}|\boldsymbol{y})$ when $N \to \infty$ (*Greenberg et al., 2019*). It is worth noting that the parametrization $\phi$, obtained at the end of the optimization procedure, serves as an amortized posterior for various choices of $\boldsymbol{x}$. Hence, for a particular observation $\boldsymbol{x}_0$, we can simply use $q_\phi(\boldsymbol{\theta}|\boldsymbol{x}_0)$ as an approximation of $p(\boldsymbol{\theta}|\boldsymbol{x}_0)$.

## µGUIDE framework

The full architecture of the proposed Bayesian framework, dubbed µGUIDE, is presented in *Figure 1*. The analysis codes underpinning the results presented here can be found on Github: https://github.com/mjallais/uGUIDE (copy archived at *Jallais, 2024*) (both CPU and GPU are supported).

µGUIDE is comprised of two modules that are optimized together to minimize the Kullback–Leibler divergence between the true posterior distribution and the estimated one for every parameters of a given forward model. The NPE module uses normalizing flows to approximate the posterior distribution, while the MLP module is used to reduce the data dimensionality and ensure fast and robust convergence of the NPE module. The following sections provide more details about our implementation of each module.

### Neural Posterior Estimator

In this study, the Sequential Neural Posterior Estimation (SNPE-C) algorithm (*Papamakarios and Murray, 2016*; *Greenberg et al., 2019*) with a single round is employed to train a neural network that directly approximates the posterior distribution. Thus, sampling from the posterior can be done by sampling from the trained neural network. Neural density estimators have the advantage of providing exact density evaluations, in contrast to Variational Autoencoders (VAEs [*Kingma and Welling, 2019*]) or generative adversarial networks (GANs [*Goodfellow et al., 2014*]), which are better suited for generating synthetic data.

The conditional probability density function approximators used in this project belong to a class of neural networks called normalizing flows (*Papamakarios et al., 2021*). These flows are invertible functions capable of transforming vectors generated from a simple base distribution (e.g. the standard multivariate Gaussian distribution) into an approximation of the true posterior distribution. An autoregressive architecture for normalizing flows is employed, implemented via the MAF (*Papamakarios et al., 2017*), which is constructed by stacking five Masked Autoencoder for Distribution Estimation (MADE) models (*Germain et al., 2015*). An explanation of how MAF and MADE work is provided in Appendix 3.

To test that the predicted posteriors for a given model are not incorrect we use posterior predictive checks (PPCs), which is described in more details in Appendix 4.

### Handling the large dimensionality of the data with MLP

As the dimensionality of the input data $\boldsymbol{x}$ grows, the complexity of the corresponding inverse problem also increases. Accurately characterizing the posterior distributions or estimating the tissue microstructure parameters becomes more challenging. As a consequence, it is often necessary to rely on a set of low-dimensional features (or summary statistics) instead of the raw data for the inference task process (*Blum et al., 2013*; *Fearnhead and Prangle, 2012*; *Papamakarios et al., 2019*). These summary statistics are features that capture the essential information within the raw data, allowing to reduce the size of the input vector. Learning a set of sufficient statistics before estimating the posterior distribution makes the inference easier and offers many benefits (see e.g. the Rao–Blackwell theorem).

A follow-up challenge lies in the choice of suitable summary statistics. For well-understood problems and data, it is possible to manually design these features using deterministic functions that condense the information contained in the raw signal into a set of handful summary statistics. Previous works, such as *Novikov et al., 2018* and *Jallais et al., 2022*, have proposed specific summary statistics for two different biophysical models. However, defining these summary statistics is difficult and often requires prior knowledge of the problem at hand. In the context of dMRI, they also rely on acquisition constraints and are model specific.

In this work, the proposed framework aims to be applicable to any forward model and be as general as possible. We therefore propose to learn the summary statistics from the high-dimensional input signals $x$ using a neural network. This neural network is referred to as an embedding neural network. The observed signals are fed into the embedding neural network, whose outputs are then passed to the neural density estimator. The parameters of the embedding network are learned together with the parameters of the neural density estimator, leading to the extraction of optimal features that minimize the uncertainty of $p(\theta|x)$. Here, we propose to use an MLP with three layers as a summary statistics extractor. The number of features $N_f$ extracted by the MLP can be either defined a priori or determined empirically during training.

## Training µGUIDE

To train µGUIDE we need couples of input vectors $x$ and corresponding ground truth values for the model parameters that we want to estimate, $\theta$. The input $x$ can be real or simulated data (e.g. dMRI signals); or a mixture of these two. We train µGUIDE by stochastically minimizing the loss function defined in *Equation (6)* using the Adam optimizer (*Kingma and Ba, 2015*) with a learning rate of $10^{-3}$ and a minibatch size of 128. We use 1 million simulations for each model, 5% of which are randomly selected to be used as a validation set. Training is stopped when the validation loss does not decrease for 30 consecutive epochs.

## Quantifying the confidence in the estimates

The full posterior distribution contains a lot of useful information about a given model parameter best estimates, uncertainty, ambiguity, and degeneracy. To summarize and easily visualize this information, we propose three measures that quantify the best estimates and the associated confidence levels, and a way to highlight degeneracy.

We start by checking whether a posterior distribution is degenerate, that is if the distribution presents multiple distinct parameter solutions, appearing as multiple local maxima (*Figure 2*). To that aim, we fit two Gaussian distributions to the obtained posterior distributions. A voxel is considered as degenerate if the derivative of the fitted Gaussian distributions changes signs more than once (i.e. multiple local maxima), and if the two Gaussian distributions are not overlapping (the distance between the two Gaussian means is inferior to the sum of their standard deviations).

For non-degenerate posterior distributions, we extract three quantities:

1. The MAP, which corresponds to the most likely parameter estimate.
2. An uncertainty measure, which quantifies the dispersion of the 50% most probable samples using the interquartile range, relative to the prior range.
3. An ambiguity measure, which measures the FWHM, in percentage with respect to the prior range.

*Figure 2* presents those measures on exemplar posterior distributions.

## Application of µGUIDE to biophysical modelling of dMRI data

We show exemplar applications of µGUIDE to three biophysical models of increasing complexity and degeneracy from the dMRI literature. For each model, we compare the fitting quality of the posterior distributions obtained using the MLP and manually defined summary statistics.

## Biophysical models of dMRI signal
### Model 1: Ball&Stick (*Behrens et al., 2003*)

This is a two-compartment model (intra- and extra-neurite space) where the dMRI signal from the brain tissue is modelled as a weighted sum, with weight $f_{in}$, of signals from water diffusing inside the neurites, approximated as sticks (i.e. cylinders of zero radius) with diffusivity $D_{in}$, and water diffusing within the extra-neurite space, approximated as Gaussian diffusion in an isotropic medium with diffusivity $D_e$. The direction of the stick is randomly sampled on a sphere. This model has the main advantage of being non-degenerate. We define the summary statistics as the direction-averaged signal (six b-shells, see section dMRI data acquisition and processing).

## Model 2: SM (*Novikov et al., 2019*)

Expanding on Model 1, this model represents the dMRI signal from the brain tissue as a weighted sum of the signal from water diffusing within the neurite space, approximated as sticks with symmetric orientation dispersion following a Watson distribution and water diffusing within the extra-neurite space, modelled as anisotropic Gaussian diffusion. The microstructure parameters of this two-compartment model are the neurite signal fraction $f$, the intra-neurite diffusivity $D_a$, the orientation dispersion index $ODI$, and the parallel and perpendicular diffusivities within the extra-neurite space $D_e^{\parallel}$ and $D_e^{\perp}$. We use the LEMONADE (*Novikov et al., 2018*) system of equations, which is based on a cumulant decomposition of the signal, to define six summary statistics.

## Model 3: extended-SANDI (*Palombo et al., 2020*)

This is a three-compartment model (intra-neurite, intra-soma, and extra-cellular space) where the dMRI signal from the brain tissue is modelled as a weighted sum of the signal from water diffusing within the neurite space, approximated as sticks with symmetric orientation dispersion following a Watson distribution; water diffusing within cell bodies (namely soma), modelled as restricted diffusion in spheres; and water diffusing within the extra-cellular space, modelled as isotropic Gaussian diffusion. The parameters of interest are the neurite signal fraction $f_n$, the intra-neurite diffusivity $D_n$, the orientation dispersion index $ODI$, the extra-cellular signal fraction $f_e$ and isotropic diffusivity $D_e$, the soma signal fraction $f_s$, and a proxy of soma radius and diffusivity $C_s$, defined as (*Jallais et al., 2022*):

$$C_s = \frac{2}{D_s\delta^2} \sum_{m=1}^{\infty} \frac{\alpha_m^{-4}}{\alpha_m^2 r_s^2 - 2} \cdot \left( 2\delta - \frac{2 + e^{-\alpha_m^2 D_s(\Delta-\delta)} - e^{-\alpha_m^2 D_s\delta} - e^{-\alpha_m^2 D_s\Delta} + e^{-\alpha_m^2 D_s(\Delta+\delta)}}{\alpha_m^2 D_s} \right) ,$$

with $r_s$ and $D_s$ the soma radius and diffusivity, respectively, and $\alpha_m$ the $m$th root of $(\alpha r_s)^{-1} J_{\frac{3}{2}}(\alpha r_s) = J_{\frac{5}{2}}(\alpha r_s)$, with $J_n(x)$ the Bessel functions of the first kind. We use the six summary statistics defined in *Jallais et al., 2022*, which are based on a high and low $b$-value signal expansion. Signal fractions follow the rule $f_n + f_s + f_e = 1$, leading to six parameters to estimate for this model.

Prior distributions $p(\boldsymbol{\theta})$ are defined as uniform distributions over biophysically plausible ranges. Signal fractions are defined within the interval $[0, 1]$, diffusivities between 0.1 and 3 µm²/ms, ODI between 0.03 and 0.95, and $C_s$ between 0.15 and 1105 µm² (which correspond to $r_s \in [1; 15]$ µm and fixed $D_s = 3$ µm²/ms).

The SM imposes the constraint $D_e^{\perp} < D_e^{\parallel}$. To generate samples uniformly distributed on the space defined by this condition, we are using two random variables $u_0$ and $u_1$, both sampled uniformly between 0 and 1, and then relate them to $D_e^{\parallel}$ and $D_e^{\perp}$ using the following equations:

$$\begin{cases} D_e^{\parallel} = \sqrt{(3.0 - 0.1)^2 \cdot u_0} + 0.1 \\ D_e^{\perp} = (D_e^{\parallel} - 0.1) \cdot u_1 + 0.1 \end{cases} \tag{7}$$

The extended-SANDI model requires for the signal fractions to sum to 1, that is $f_n + f_s + f_e = 1$. To uniformly cover the simplex $f_n + f_s + f_e = 1$, we define two new parameters $k_1$ and $k_2$, uniformly sampled between 0 and 1, and use the following equations to get the corresponding signal fractions:

$$\begin{cases} f_n = k_2\sqrt{k_1} \\ f_s = (1 - k_2)\sqrt{k_1} \\ f_e = 1 - \sqrt{k_1} \end{cases} \tag{8}$$

To ensure comparability of results, we extract the same number of features $N_f$ using the MLP as the number of summary statistics for each model. We therefore use $N_f = 6$ for the Ball&Stick, the SM, and the extended-SANDI models. Although the number of features predicted by the MLP is fixed to $N_f = 6$ for the three models, the characteristics of these six features can be very different, depending on the chosen forward model and the available data (see Appendix 1). Training the MLP together with the NPE module allows to maximize inference performance in terms of accuracy and precision.

## Validation in numerical simulations

We start by validating the proposed method using PPCs and simulated signals from Model 2 (see more details in Appendix 4). Since PPC alone does not guarantee the correctness of the estimated posteriors, we further validated the obtained posterior distributions comparing them with the AMWG MCMC (*Roberts and Rosenthal, 2009*). We generated simulations following the same acquisition protocol as the real data (see section dMRI data acquisition and processing), added Gaussian noise to the real and imaginary parts of the simulated signal with a signal-to-noise ratio (SNR) of 50, and then used the magnitude of this noisy complex signal for our experiments. We then estimated the posterior distributions using both µGUIDE and the MCMC method implemented in the MDT toolbox (*Harms and Roebroeck, 2018*). We initialized the sampling using an MLE. We sampled 15,200 samples from the distribution, the first 200 ones being used as burn-in, and no thinning. Similarly, we sampled 15,000 samples from the estimated posterior distributions using µGUIDE.

Then, we show that µGUIDE can be applicable to any model. We use Models 1 and 3 as examples of simpler (and non-degenerate) and more complex (and degenerate) models than Model 2, respectively.

We compared the proposed framework to a state-of-the-art method for posterior estimation (*Jallais et al., 2022*). This method relies on manually defined summary statistics, while µGUIDE automatically extracts some features using an embedded neural network. µGUIDE was trained directly on noisy simulations. The manually defined summary statistics were extracted from these simulated noisy signals and then used as training dataset for an MAF, similar to *Jallais et al., 2022*.

Finally, we used µGUIDE to highlight degeneracy in all the models. While the complexity of the models increases, more degeneracy can be found. The degeneracy is inherent to the model definition, and is not induced by the noise. µGUIDE allows to highlight those degeneracies and quantify the confidence in the obtained estimates.

The training was performed on $N = 10^6$ numerical simulations for each model, computed using the MISST package (*Ianuş et al., 2017*) and random combinations of the model parameters, each uniformly sampled from the previously defined ranges, with the addition of Rician distributed noise with SNR equivalent to the experimental data, that is 50.

## dMRI data acquisition and processing

We applied µGUIDE to dMRI data collected from two participants: a healthy volunteer from the WAND dataset (*McNabb et al., 2024*) and an age-matched participant with epilepsy, acquired with the same protocol used for the MICRA dataset (*Koller et al., 2021*). The MRI data from the healthy volunteer used in this work are part of a previously published dataset, publicly available at https://doi.gin.g-node.org/10.12751/g-node.5mv3bf/. We do not have the authorization to share the epileptic patient data. Data were acquired on a Connectome 3T scanner using a single-shot spin-echo, echo-planar imaging sequence with *b*-values = [200, 500, 1200, 2400, 4000, 6000] s mm⁻², [20, 20, 30, 61, 61, 61] uniformly distributed directions, respectively, and 13 non-diffusion-weighted images at 2 mm isotropic resolution. TR was set to 3000 ms, TE to 59 ms, and the diffusion gradient duration and separation to 7 ms and 24 ms, respectively. Short diffusion times and TE were achieved thanks to the Connectom gradients, allowing to enhance the SNR and sensitivity to small water displacements (*Jones et al., 2018*; *Setsompop et al., 2013*). We considered the noise as Rician with an SNR of 50 for both subjects.

Data were preprocessed using a combination of in-house pipelines and tools from the FSL (*Andersson et al., 2003*; *Andersson and Sotiropoulos, 2016*; *Smith, 2002*; *Smith et al., 2004*) and MRTrix3 (*Tournier et al., 2019*) software packages. The preprocessing steps included brain extraction (*Smith, 2002*), denoising (*Cordero-Grande et al., 2019*; *Veraart et al., 2016*), drift correction (*Vos et al., 2017*; *Sairanen et al., 2018*), susceptibility-induced distortions (*Andersson et al., 2003*; *Smith et al., 2004*), motion and eddy current correction (*Andersson and Sotiropoulos, 2016*), correction for gradient non-linearity distortions (*Glasser et al., 2013*), and Gibbs ringing artefacts correction (*Kellner et al., 2016*).

## dMRI data analysis

Diffusion signals were first normalized by the mean non-diffusion-weighted signals acquired for each voxel. Each voxel was then estimated in parallel using the µGUIDE framework. For each observed

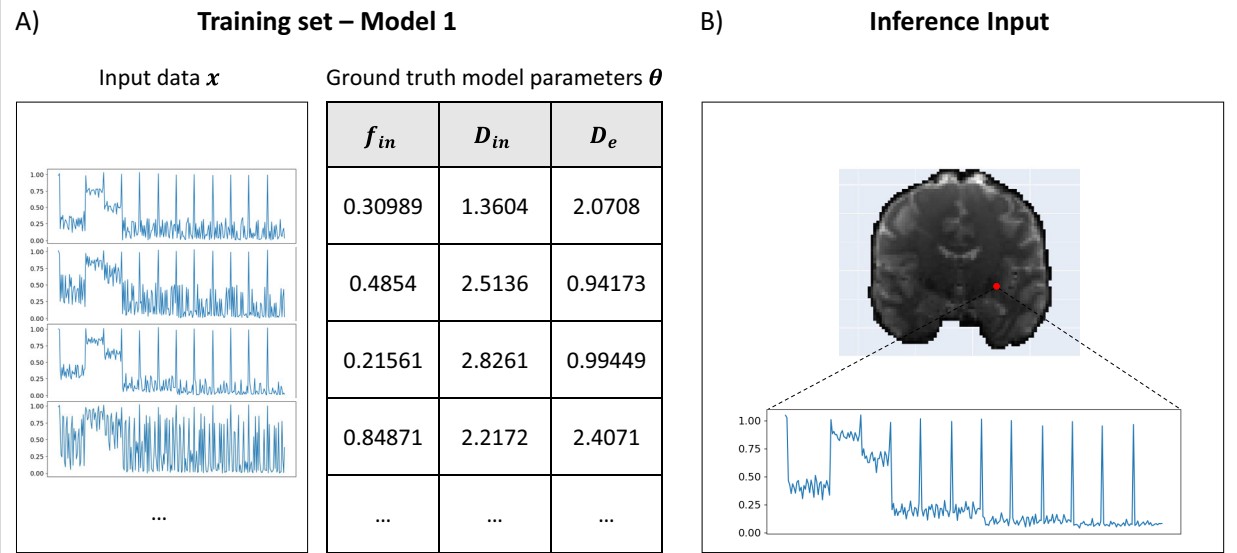

**Figure 8.** Example training set and input signals for µGUIDE. (**A**) Examples of input synthetic data vectors and corresponding ground truth model parameters used in the training set of Model 1 (Ball&Stick). (**B**) Example of input measured signals from a voxel in a healthy participant, used for inference.

signal $x_v$ (i.e. for each voxel), we drew 50,000 samples via rejection sampling from $q_\phi(\theta_i|x_v)$ for each model parameter $\theta_i$, allowing to retrieve the full posterior distributions. If a posterior distribution was not deemed degenerate, the MAP, uncertainty, and ambiguity measures were extracted from the posterior distributions.

The manually defined summary statistics of the SM are defined using a cumulant expansion, which is only valid for small $b$-values. We therefore only used the $b \leq 2500$ s mm$^{-2}$ data for this model. In order to obtain comparable results, we restricted the application of µGUIDE to this range of $b$-values as well. An extra b-shell ($b$-value = 5000 s mm$^{-2}$; 61 directions) was interpolated using *mapl* (*Fick et al., 2016*) for the extended-SANDI model when using the method developed by *Jallais et al., 2022* based on summary statistics.

The training of µGUIDE was performed as described in section Validation in numerical simulations and an example of training dataset and input signal vector is provided in *Figure 8*.

All the computations were performed both on CPU and GPU (NVIDIA GeForce RTX 4090).

## Acknowledgements

This work, Maëliss Jallais and Marco Palombo are supported by UKRI Future Leaders Fellowship (MR/T020296/2). We are thankful to Dr. Dmitri Sastin and Dr. Khalid Hamandi for sharing their dataset from a participant with epilepsy, and to Dr. Carolyn McNabb, Dr. Eirini Messaritaki, and Dr. Pedro Luque Laguna for preprocessing the data of the healthy participant from the WAND data. The WAND data were acquired at the UK National Facility for In Vivo MR Imaging of Human Tissue Microstructure funded by the EPSRC (grant EP/M029778/1) and The Wolfson Foundation, and supported by a Wellcome Trust Investigator Award (096646/Z/11/Z) and a Wellcome Trust Strategic Award (104943/Z/14/Z). The WAND data are available at https://doi.gin.g-node.org/10.12751/g-node.5mv3bf/.

# Additional information

## Funding

| Funder | Grant reference number | Author |
|---|---|---|
| UK Research and Innovation | Future Leaders Fellowship MR/T020296/2 | Marco Palombo |

The funders had no role in study design, data collection, and interpretation, or the decision to submit the work for publication.

## Author contributions

Maëliss Jallais, Conceptualization, Data curation, Software, Formal analysis, Validation, Investigation, Visualization, Methodology, Writing – original draft, Writing – review and editing; Marco Palombo, Conceptualization, Resources, Supervision, Funding acquisition, Validation, Visualization, Methodology, Writing – original draft, Project administration, Writing – review and editing

## Author ORCIDs

Maëliss Jallais ⓘ https://orcid.org/0000-0001-5939-388X
Marco Palombo ⓘ https://orcid.org/0000-0003-4892-7967

Reviewer #1 (Public review): https://doi.org/10.7554/eLife.101069.3.sa1
Reviewer #2 (Public review): https://doi.org/10.7554/eLife.101069.3.sa2
Author response https://doi.org/10.7554/eLife.101069.3.sa3

# Additional files

## Supplementary files

• MDAR checklist

## Data availability

The current manuscript is a computational study, so no new data have been generated for this manuscript. The MRI data used in this work are part of a previously published dataset, publicly available at https://doi.gin.g-node.org/10.12751/g-node.5mv3bf/. We do not have the authorization to share the epileptic patient data. The analysis codes underpinning the results presented here can be found on Github: https://github.com/mjallais/uGUIDE, (copy archived at *Jallais, 2024*).

The following previously published dataset was used:

| Author(s) | Year | Dataset title | Dataset URL | Database and Identifier |
|---|---|---|---|---|
| McNabb CB, Driver ID, Hyde V, Hughes G, Chandler HL, Thomas H, Allen C, Messaritaki E, Hodgetts CJ, Hedge C, Engel M, Standen SF, Morgan EL, Stylianopoulou E, Manalova S, Reed L, Drakesmith M, Germuska M, Shaw AD, Mueller L, Rossiter H, Davies-Jenkins CW, Lancaster T, Evans CJ, Owen D, Perry G, Kusmia S, Lambe E, Partridge AM, Cooper A, Hobden P, Lu H, Graham KS, Lawrence AD, Wise RG, Walters JTR, Sumner P, Singh KD, Jones DK | 2024 | The Welsh Advanced Neuroimaging Database (WAND) | https://doi.gin.g-node.org/10.12751/g-node.5mv3bf/ | G-Node, 10.12751/g-node.5mv3bf |

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

## Appendix 1

### Correlation analysis between features extracted by µGUIDE and manually defined summary statistics

*Appendix 1—figure 1* correlation presents the correlation matrices obtained from the correlation between the MLP-extracted features from µGUIDE and the manually defined summary statistics defined in section Application of µGUIDE to biophysical modelling of dMRI data, considering noisy simulations (SNR = 50). For each model, at least one feature extracted by µGUIDE is not or weakly correlated with the summary statistics. Additional information, not contained in the summary statistics, is extracted by the MLP from the input signal, leading to reduced bias, uncertainty and ambiguity in the parameter estimates (see *Figure 4*).

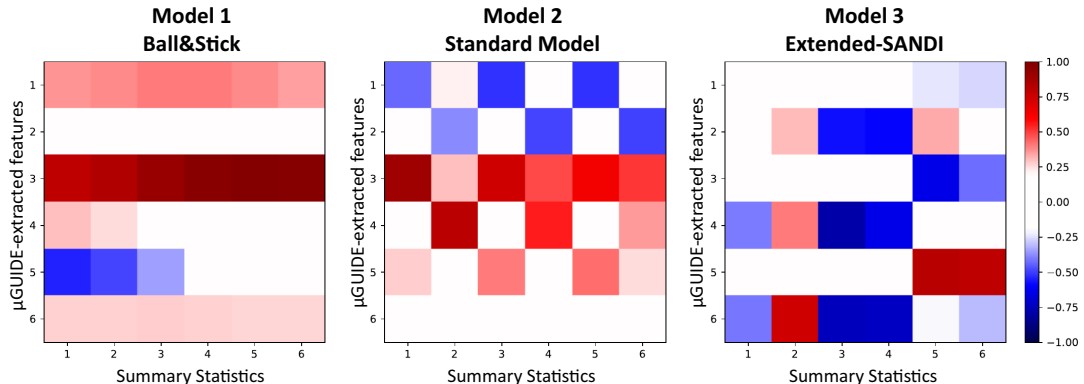

**Appendix 1—figure 1.** Correlation matrices between features extracted by the Multi-Layer Perceptron (MLP) in µGUIDE and manually defined summary features for the three models.

# Appendix 2

## Impact of noise on the posterior distributions

Noise in the signal impacts the fitting quality of a biophysical model. *Appendix 2—figure 1A* shows example posterior distributions for one combination of Model 2 parameters, with varying noise levels (no noise, $SNR = 50$, and $SNR = 25$). *Appendix 2—figure 1B* presents uncertainties values obtained on 1000 simulations with varying SNRs. We observe that, as the SNR reduces (i.e. as the noise increases), uncertainty increases. Noise in the signal contributes to irreducible variance. The confidence in the estimates therefore reduces as the noise level increases.

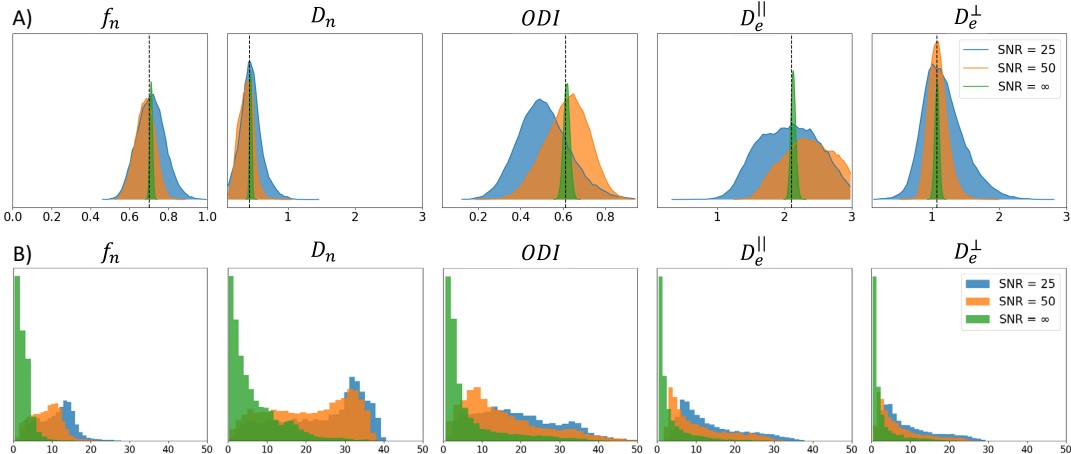

**Appendix 2—figure 1.** SNR uncertainty comparison between signals with different noise levels: no noise, $SNR = 50$, and $SNR = 25$ using Model 2. (**A**) Posterior distributions obtained on one example parameter combination (vertical black dashed line) with the three noise levels. (**B**) Histogram of the uncertainty obtained for 1000 signals with different noise levels (in %). Similar ground truths are used for each noise level.

# Appendix 3

## Masked Autoregressive Flows

The conditional probability density function approximators used in this project belong to a class of neural networks called normalizing flows (NFs [*Papamakarios et al., 2021*]). NFs provide a general way of transforming complex probability distributions over continuous random variables into simple base distributions $p(z)$ (such as normal distributions) through a chain of invertible and differentiable transformations $f_\phi$. By applying the change of variable formula, the target distribution $q_\phi(\theta|x)$ can be written as:

$$q_\phi(\theta \mid x) = p\left(f_\phi(\theta; x)\right) \left|\det J_{f_\phi}(\theta; x)\right|, \tag{9}$$

where $z = f_\phi(\theta; x)$ is invertible and differentiable (i.e. a diffeomorphism), and $J_{f_\phi}(\theta; x)$ is the Jacobian of $f_\phi(\theta; x)$. The forward direction allows for density evaluation, that is learning the mapping between the target and the base distributions, that is learning the parameters $\phi$. The inverse direction allows to estimate a density estimator $q_\phi(\theta \mid x_0)$ by sampling points $z$ from the base distribution and applying the inverse transform $f_\phi^{-1}(z; x_0)$. A main requirement is that the flow needs to be expressive enough to approximate any arbitrarily highly complex distribution. An interesting property of diffeomorphisms is that they are closed under composition, which means that a composition of $K$ diffeomorphisms $f = f_1 \circ f_2 \circ \cdots \circ f_K$ is also a diffeomorphism, and the Jacobian determinant is the product of the determinant of each component. Combining multiple transformations allows to increase the expressivity of the general flow. We obtain:

$$
\begin{aligned}
q_\phi(\theta \mid x) &= p\left(f_\phi(\theta; x)\right) \log \left|\det \prod_{k=1}^{K} J_{f_k}\left(\theta_i; x_i\right)\right| \\
&= p\left(f_\phi(\theta; x)\right) \sum_{k=1}^{K} \log \left|\det J_{f_k}\left(\theta_i; x_i\right)\right|
\end{aligned}
\tag{10}
$$

Flows need to be flexible and expressive enough to model any desired distribution but also need to be computationally efficient, that is, computing the associated Jacobian determinants need to be tractable and efficient. Among a number of proposed architectures such as mixture density networks (*Bishop, 1994*) or neural spline flows (*Durkan et al., 2019*), we focused on MAFs (*Papamakarios et al., 2017*), which has shown state-of-the-art performance as well as the ability to estimate multi-modal posterior distributions (*Gonçalves et al., 2020*; *Papamakarios et al., 2021*; *Patron et al., 2022*).

Autoregressive flows are universal approximators and have the form $z_i' = \tau(z_i; h_i)$ where $h_i = c_i(z_{<i})$ (*Papamakarios et al., 2021*). $\tau$ is termed the transformer and is a strictly monotonic function parametrized by $h_i$, and $c_i$ the $i$th conditioner. Each $h_i$ and therefore each $z_i'$ can be computed independently in parallel, helping to keep a low computation time. The conditioner constraints each output to depend only on variables with dimension indices less than $i$, which makes the Jacobian of the flow lower diagonal. Its determinant can then be obtained easily as the product of its diagonal elements. To efficiently implement the conditioner, this method relies on the MADE (*Germain et al., 2015*) architecture. To create a neural network that obey the autoregressive structure of the conditioner, a fully connected feedforward neural network is multiplied to binary masks, which removes some connections by assigning them a weigh of 0. The binary masks can easily be obtained by following a few simple steps (see *Appendix 3—figure 1* for an illustration):

1. Label the input and output nodes between 1 and $D$, $D$ being the dimension of the input vector $z$.
2. Randomly assign each hidden unit a number between 1 and $D - 1$, which indicates the number of inputs it will be connected to.
3. For each hidden layer, connect the hidden units to units with inferior or equal labels.
4. Connect output units to units with strictly inferior labels.

For μGUIDE's implementation, we use a combination of five MADEs. As a result, the MAF architecture only needs a single forward pass through the flow and, combined with the low-cost computation of the determinant, allows for fast training and evaluation of the posterior distributions.

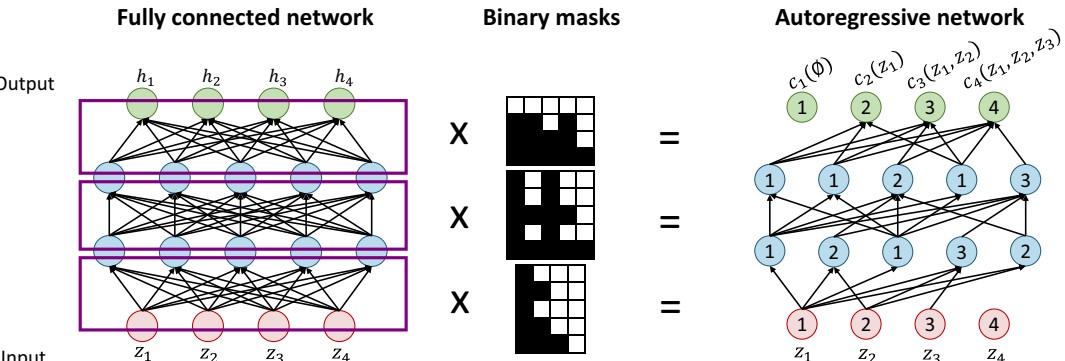

**Appendix 3—figure 1.** Schematic of Masked Autoencoder for Distribution Estimation (MADE) autoregressive network construction.

## Appendix 4

### Posterior predictive checks

PPCs are a common safety check to verify inference is not wrong. The idea is to compare input signals with generated signals from samples drawn from the posterior distributions. If the inference is correct, the generated signals should look similar to the input signal. We performed the following steps:

- Sample $\theta_i\theta_i$ from the prior distribution: $\theta_i \sim p(\theta)$
- Generate the corresponding signals using the forward model: $x_i = \mathcal{M}(\theta_i)$
- Perform the inference and estimate the posterior distributions $p(\theta_i|x_i)$
- Sample $N_{PP}$ samples $\theta_{i,s}$ from $p(\theta_i|x_i)$
- Reconstruct the signals from the sampled $\theta_{i,s}$ using the forward model: $x_{i,s} = \mathcal{M}(\theta_{i,s})$
- Compare the obtained $x_{i,s}$ with $x_i$.

*Appendix 4—figure 1* presents results on Model 2 (SM), on both noise-free and noisy signals (Rician noise with SNR =50) for $N = 10$ random combinations of model parameters, and $N_{PP} = 100$. As dMRI data have a high dimensionality, we report the direction-average signal. Plain lines show the signals $x_i$, and the shaded areas correspond to the area in which the corresponding $x_{i,s}$ fall. $x_i$ lie within the support of $x_{i,s}$, indicating the inference is not wrong. Note that the support of $x_{i,s}$ is bigger for noisy simulations, reflecting the wider posterior distributions obtained from the inference.

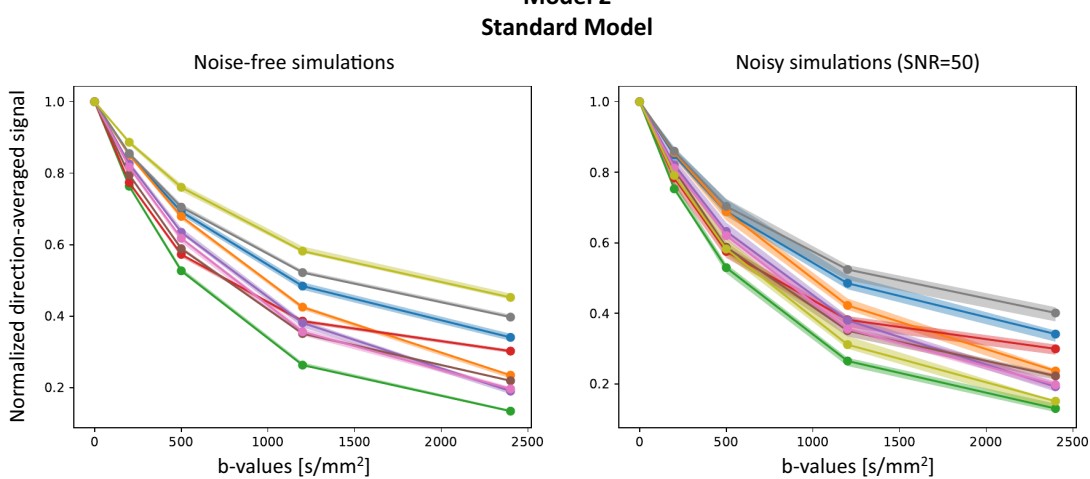

**Appendix 4—figure 1.** Posterior predictive checks. Comparison between signals $x_i$ generated using random parameter combinations and their reconstructions using samples from $p(\theta_i|x_i)$.

