## [Editor Report · eLife Assessment]

The authors proposed an **important** novel deep-learning framework to estimate posterior distributions of tissue microstructure parameters. The method shows superior performance to conventional Bayesian approaches and there is **convincing** evidence for generalizing the method to use data from different protocol acquisitions and work with models of varying complexity.

---

## [Referee Report · Reviewer #1 (Public review)]

The authors proposed a framework to estimate the posterior distribution of parameters in biophysical models. The framework has two modules: the first MLP module is used to reduce data dimensionality and the second NPE module is used to approximate the desired posterior distribution. The results show that the MLP module can capture additional information compared to manually defined summary statistics. By using the NPE module, the repetitive evaluation of the forward model is avoided, thus making the framework computationally efficient. The results show the framework has promise in identifying degeneracy. This is an interesting work.

Comment on revised version:

The authors have addressed all the raised concerns and made appropriate modifications to the manuscript. The changes have improved the clarity, methodology, and overall quality of the paper. Given these improvements, I believe the paper now meets the standards for publication in this journal.

---

## [Referee Report · Reviewer #2 (Public review)]

Summary:

The authors improve the work of Jallais et al. (2022) by including a novel module capable of automatically learning feature selection from different acquisition protocols inside a supervised learning framework. Combining the module above with an estimation framework for estimating the posterior distribution of model parameters, they obtain rich probabilistic information (uncertainty and degeneracy) on the parameters in a reasonable computation time.

The main contributions of the work are:

(1) The whole framework allows the user to avoid manually defining summary statistics, which may be slow and tedious and affect the quality of the results.

(2) The authors tested the proposal by tackling three different biophysical models for brain tissue and using data with characteristics commonly used by the diffusion-MR-microstructure research community.

(3) The authors validated their method well with the state-of-the-art.

(4) The methodology allows the quantification of the inherent model's degeneration and how it increases with strong noise.

The authors showed the utility of their proposal by computing complex parameter descriptors automatically in an achievable time for three different and relevant biophysical models.

Importantly, this proposal promotes tackling, analyzing, and considering the degenerated nature of the most used models in brain microstructure estimation.

---

## [Author Response]

The following is the authors’ response to the original reviews.

**Reviewer #1 (Public Review):**
The authors proposed a framework to estimate the posterior distribution of parameters in biophysical models. The framework has two modules: the first MLP module is used to reduce data dimensionality and the second NPE module is used to approximate the desired posterior distribution. The results show that the MLP module can capture additional information compared to manually defined summary statistics. By using the NPE module, the repetitive evaluation of the forward model is avoided, thus making the framework computationally efficient. The results show the framework has promise in identifying degeneracy. This is an interesting work.

We thank the reviewer for the positive comments made on our manuscript.

**Reviewer #1 (Recommendations For The Authors):**
I have some minor comments.(1) The uGUIDE framework has two modules, MLP and NPE. Why are the two modules trained jointly? The MLP module is used to reduce data dimensionality. Given that the number of features for different models is all fixed to 6, why does one need different MLPs? This module should, in principle, be general-purpose and independent of the model used.

The MLP must be trained together with the NPE module to maximise inference performance in terms of accuracy and precision. Although the number of features predicted by the MLP was fixed to six, the characteristics of these six features can be very different, depending on the chosen forward model and the available data, as we showed in Appendix 1 Figure 1. Training the MLP independently of the NPE would result in suboptimal performance of µGUIDE, with potentially higher bias and variance of the predicted posterior distributions. We have now added these considerations in the Methods section.

(2) The authors mentioned at L463 that all the 3 models use 6 features. From L445 to L447, it seems model 3 has 7 unknown parameters. How can one use 6 features to estimate 7 unknowns?

Thank you for pointing out the lack of clarity regarding the parameters to estimate in this section. Model 3 is a three-compartment model, whose parameters of interest are the signal fraction and diffusivity from water diffusing in the neurite space (*fn* and *Dn*), the neurites orientation dispersion index (*ODI*), the signal fraction in cell bodies (*fs*), a proxy to soma radius and diffusivity (*Cs*), and the signal fraction and diffusivity in the extracellular space (*fe* and *De*). The signal fractions are constrained by the relationship *fn + fs + fe = 1*, hence _fe i_s calculated from the estimated *fn* and *fs*. This leaves us with 6 parameters to estimate: *fn, Dn, ODI, fs, Cs, De.* We clarified it in the revised version of the paper.

(3) L471, Rician noise is not a proper term. Rician distribution is the distribution of pixel intensities observed in the presence of noise. And Rician distribution is the result of magnitude reconstruction. See "Noise in magnitude magnetic resonance images" published in 2008. I assume that real-valued Gaussian noise is added to simulated data.

We apologize for the confusion. We added Gaussian noise to the real and imaginary parts of the simulated signals and then used the magnitude of this noisy complex signal for our experiments. We rephrased the sentence for more clarity.

(4) L475, why thinning is not used in MCMC? In figure 3, the MCMC results are more biased than uGUIDE, is it related to no thinning in MCMC?

We followed the recommendations by Harms et al. (2018) for the MCMC experiments. They analysed the impact of thinning (among other parameters) on the estimated posterior distributions. Their findings indicate that thinning is unnecessary and inefficient, and they recommend using more samples instead. For further details, we refer the reviewer to their publication, along with the theoretical works they cite. We have now added this note in the Methods section.

(5) Did the authors try model-fitting methods with different initializations to get a distribution of the parameters? Like the paper "Degeneracy in model parameter estimation for multi‐compartmental diffusion in neuronal tissue". For the in vivo data, it is informative to see the model-fitting results.

No, we did not try model-fitting methods with different initializations because such methods provide only a partial description of the solution landscape, which can be interpreted as a partial posterior distribution. Although this approach can help to highlight the problem of degeneracy, it does not provide a complete description of all potential solutions. In contrast, MCMC estimates the full posterior distribution, offering a more accurate and precise characterization of degeneracies and uncertainties compared to model-fitting methods with varying initializations. Hence, we decided to use MCMC as benchmark. We have now added these considerations to the Discussion section.

**Reviewer #2 (Public Review):**
Summary:The authors improve the work of Jallais et al. (2022) by including a novel module capable of automatically learning feature selection from different acquisition protocols inside a supervised learning framework. Combining the module above with an estimation framework for estimating the posterior distribution of model parameters, they obtain rich probabilistic information (uncertainty and degeneracy) on the parameters in a reasonable computation time.The main contributions of the work are:(1) The whole framework allows the user to avoid manually defining summary statistics, which may be slow and tedious and affect the quality of the results.(2) The authors tested the proposal by tackling three different biophysical models for brain tissue and using data with characteristics commonly used by the diffusion-MRmicrostructure research community.(3) The authors validated their method well with the state-of-the-art.The main weakness is:(1) The methodology was tested only on scenarios with a signal-to-noise ratio (SNR) equal to 50. It is interesting to show results with lower SNR and without noise that the method can detect the model's inherent degenerations and how the degeneration increases when strong noise is present. I suggest expanding the Figure in Appendix 1 to include this information.The authors showed the utility of their proposal by computing complex parameter descriptors automatically in an achievable time for three different and relevant biophysical models.

Importantly, this proposal promotes tackling, analysing, and considering the degenerated nature of the most used models in brain microstructure estimation.

We thank the reviewer for these positive remarks.

Concerning the main weakness highlighted by the reviewer: In our submitted work, we presented results both without noise and with a signal-to-noise ratio (SNR) equal to 50 (similar to the SNR in the experimental data analysed). Figure 5 shows exemplar posterior distributions obtained in a noise-free scenario, and Table 1 reports the number of degeneracies for each model on 10000 noise-free simulations. These results highlight that the presence of degeneracies is inherent to the model definition. Figures 3, 6 and 7 present results considering an SNR of 50. We acknowledge that results with lower SNR have not been included in the initial submission. To address this, we added a figure in the appendix illustrating the impact of noise on the posterior distributions. Specifically, Figure 1A of Appendix 2 shows posterior distributions estimated from signals generated using an exemplar set of model parameters with varying noise levels

(no noise, SNR=50 and SNR=25). Figure 1B presents uncertainties values obtained on 1000 simulations for each noise level. We observe that, as the SNR reduces, uncertainty increases. Noise in the signal contributes to irreducible variance. The confidence in the estimates therefore reduces as the noise level increases.

**Reviewer #2 (Recommendations For The Authors):**
Some suggestions:Panel A of Figure 2 may deserve a better explanation in the Figure's caption.

We agree that the description of panel A of figure 2 was succinct and added more explanation in the figure’s caption.

The caption of Figure 3 should mention that the panel's titles are the parameters of the used biophysical models.

We added in the caption of figure 3 that the names of the model parameters are indicated in the titles of the panels. We apologise for the confusion it may have created.

In equation (3), the authors should indicate the summation index.

We apologise for not putting the summation index in equation 3. We added it in the revised version.

In line 474, the authors should discuss if the systematic use of the maximum likelihood estimator as an initializer for the sampling does not bias the computed results.

Concerning the MCMC estimations, we followed the recommendations from Harms et al. (2018). They investigated the use of starting from the maximum likelihood estimator (MLE). They concluded that starting from the MLE allows to start in the stationary distribution of the Markov chain, removing the need for some burn-in. Additionally, they showed that initializing the sampling from the MLE has the advantage of removing salt- and pepper-like noise from the resulting mean and standard deviation maps. We have now added this note in the Methods section.